# Analysis of Human and Microbial Salivary Proteomes in Children Offers Insights on the Molecular Pathogenesis of Molar-Incisor Hypomineralization

**DOI:** 10.3390/biomedicines10092061

**Published:** 2022-08-24

**Authors:** Eftychia Pappa, Heleni Vastardis, Manousos Makridakis, Jerome Zoidakis, Konstantinos Vougas, George Stamatakis, Martina Samiotaki, Christos Rahiotis

**Affiliations:** 1Department of Operative Dentistry, School of Dentistry, National and Kapodistrian University of Athens, 115 27 Athens, Greece; 2Department of Orthodontics, School of Dentistry, National and Kapodistrian University of Athens, 115 27 Athens, Greece; 3Biotechnology Division, Systems Biology Center, Biomedical Research Foundation of the Academy of Athens, 115 27 Athens, Greece; 4Proteomics Laboratory, Biomedical Research Foundation of the Academy of Athens, 115 27 Athens, Greece; 5Institute for Bioinnovation, Biomedical Sciences Research Center “Alexander Fleming”, 166 72 Vari, Greece

**Keywords:** human, molar incisor hypomineralization (MIH), saliva, proteomics, biomarker, microbiome

## Abstract

Molar incisor hypomineralization is a complex developmental enamel defect that affects the permanent dentition of children with significant functional and aesthetic implications. Saliva is an ideal diagnostic tool and ensures patients’ compliance by diminishing the discomfort especially in pediatric population. Lately, salivary proteome analysis has progressively evolved in various biomedical disciplines. As changes in saliva composition are associated with oral diseases, it is reasonable to assume that the saliva proteome of MIH-affected children might be altered compared to healthy children. This study analyzed the human and microbial salivary proteome in children with MIH in order to identify salivary markers indicative of the pathology. The conducted proteomic analysis generated a comprehensive dataset comprising a total of 1515 high confidence identifications and revealed a clear discrimination between the two groups. Statistical comparison identified 142 differentially expressed proteins, while the pathway analysis indicated deregulation of inflammation, immune response mechanisms, and defense response to bacteria in MIH patients. Bacterial proteome analysis showed a lower diversity for the microbial species, which highlights the dysbiotic environment established in the MIH pathology.

## 1. Introduction

Molar incisor hypomineralization (MIH) is a form of developmental enamel defect (DDE) that affects both function and aesthetics of teeth. It was first described in 2001 by Weerheijm et al. [1]. Since then, numerous studies have been dedicated to its prevalence and risk factors. Meanwhile, the increasing number of recent systematic reviews underlines the growing interest in this condition [2,3]. In MIH, permanent first molars and incisors are affected, with variable prevalence depending on the country, region, or age group studied. Two recent systematic reviews and meta-analyses revealed high global mean prevalence; the first one reaching 14.2% (8.1–21.1%) [4,5] and the second going up to 12.9% (11.7–14.3%) [6].

The 2010 EAPD’s ‘policy document’ stated, ‘It is likely that MIH is not caused by one specific factor. Instead, several harmful agents/conditions may act together and increase the risk of MIH occurring additively or synergistically’ [7]. The evidence confirms that certain systemic and genetic factors act synergistically to induce enamel hypomineralization. In addition, the duration, strength, and timing of these factors may be responsible for the varied clinical characteristics of the defect. More than 30 systemic etiological hypotheses have been identified over the last 10 years; some are well established, and others are more contemporary [8].

However, from etiological and healthcare perspectives, it is a complex problem with patient and clinician implications. In MIH, the disturbance occurs in the enamel’s transitional and/or maturation stage [9]. It is now widely accepted that hypomineralization may result when the resorptive potential of ameloblasts is disturbed, and inhibition of proteolytic enzymes occurs. These alterations result in protein retention, particularly amelogenin, interfering with crystal growth and enamel maturation [10]. In a recent publication, the mineralization-poisoning model was presented, paving new ways to understanding the pathogenesis of MIH [11]. In contradiction to the etiological dogma about systemic injury to ameloblasts, biochemical analysis of chalky enamel opacities has now shifted attention to direct disruption of enamel hardening by serum albumin [10]. MIH-affected enamel shows lower mineral density and higher protein content, leading to increased porosity, especially in yellow and brown lesions. An important question is whether the increased porosity on these surfaces favors the attachment of specific microbial species and affects the composition of the biofilm [11].

Saliva surrounds the hard and soft tissues of the oral cavity, comprising organic and inorganic components. It is identified as functionally equivalent to serum, reflecting the body’s physiological state, including hormonal, emotional, nutritional, and metabolic alterations [12]. Saliva’s collection is an easy, non-invasive, effortless, chair-side procedure requiring no special equipment. It ensures patients’ compliance by diminishing the discomfort often associated with blood and urine collection [13]. It is an ideal diagnostic tool for studies conducted on special populations such as children, anxious, handicapped, or elderly patients [14]. As the primary host-associated factor, saliva plays an essential role in the dynamic equilibrium in the oral cavity.

In the last decade, salivary proteomics have been applied in order to unveil pathological mechanisms on protein expression level for various oral and systemic conditions [15]. As changes in saliva composition are associated with oral pathologies, it is reasonable to assume that the saliva proteome of MIH-affected children might be altered compared to healthy children. While several studies have attempted to identify proteomic profiles in oral diseases [16], there are only scarce data in the literature about the proteome in MIH pathology, indicating differential expression of proteins in MIH patients’ saliva [17].

Saliva’s diagnostic potential to detect hypomineralized lesions long before tooth eruption could ensure timely clinician’s intervention for prevention of enamel breakdown after application of mastication forces. The current study aims to characterize the human and microbial proteome in saliva samples collected from adolescents with MIH and matched controls, in order to identify salivary markers indicative of the pathology.

## 2. Materials and Methods

### 2.1. Study Design

According to the Declaration of Helsinki, this study protocol and written consent forms were approved by the Medical Ethics Committee of the School of Dentistry of the National and Kapodistrian University of Athens (ethical approval code 462/07.04.2021). All of the experimental methods were performed following the relevant guidelines and regulations. The protocol was explained to both parents and children, and written informed consent to participate in the study was obtained from a parent. A total of 20 subjects participated in the study and were allocated into two groups. The MIH group comprised 10 children with MIH defects, and the control group (Ctrl) comprised 10 healthy subjects who were sex-and-aged-matched accordingly. The control group was established by matching each participant, who did not have any systemic disease nor receive any medication, to each of the adolescents with MIH. Besides age and gender, the matching criteria were city of residence, fluoride exposure, social background (based on parental education level), and oral hygiene routine. A questionnaire analysis determined caries risk factors for all the participants in the study, so as to eliminate differences between groups with regard to dietary habits, oral hygiene, dental visits, fluoride intake, and social background. A specialized dentist examined all subjects in the Orthodontics Department at the Dental School of National and Kapodistrian University of Athens. Diagnosis of MIH was based upon European Academy of Pediatric Dentistry guidelines and criteria [18]. Participants were additionally examined for the presence of oral inflammation, and both PI and GI indexes were recorded. The following protocol was applied to effectively control potential sources of variability in the composition of saliva.

### 2.2. Standardized Sample Collection

Before the examination day, participants were advised not to eat or drink one hour before their scheduled appointment. Unstimulated whole saliva was collected from all participants. All saliva samples were collected between 10:00 AM and 12:00 PM, to minimize any inter-individual variation of saliva composition associated with circadian rhythms [19]. The samples were discarded if the subject became stressed or began to cry. Additionally, the plaque index and gingival index were recorded. Subjects with oral inflammation and subjects receiving antimicrobial therapy were excluded from the study [20]. At the moment of saliva collection, a specialized dentist examined all participating subjects, using as the exclusion criteria the presence of gingivitis or any clinical signs of oral inflammation. Unstimulated whole saliva was collected as it flowed into the anterior floor of the mouth via passive drooling to a plastic tube. Collection tubes were stored on ice during the examination to prevent proteolytic degradation of salivary proteins.

### 2.3. Sample Processing

Samples (3 mL) were added in 5 mL of 10% SDS and incubated at 56 °C o/n. Next, 5% SDS and overnight heating at 56 °C was performed as a precaution mean to neutralize possible COVID-19 presence in the samples [21,22]. TCA precipitation was performed with 10% *w*/*v* TCA o/n at −20 °C. Samples were centrifuged at 10,000 rcf, 4C for 10 min. The protein pellet was washed with ice-cold acetone (10,000 rcf, 4C, 10 min). The supernatant was discarded, and the protein pellet was air-dried for 10 min. The protein pellet of each sample was resuspended in 300 μL lysis buffer (7M Urea, 2M Thiourea, 4% CHAPS, 1% DTE, 3.6% Protease Inhibitors). Water bath sonication was performed for 10 min to enhance the solubilization process. Samples were subjected to buffer exchange with 50 mM ammonium bicarbonate followed by concentration with 3 kDa MWCO Amicon filters. The final volume of the concentrated samples was 30 μL. Protein concentration was defined by the Bradford Assay. Samples were prepared with the Filter Aided Sample Preparation (FASP) method as previously reported [23] with minor modifications [24]. Then, 200 μg of each sample were loaded into the filter. For the samples with protein content lower than 200 μg, the entire protein amount was loaded into the filter. Alkylation was performed with 0.05 M iodoacetamide in 50 mM ammonium bicarbonate for 20 min, RT in the dark. Samples were centrifuged (14,000 rcf, 10 min, RT) and washed twice with 50 mM ammonium bicarbonate by centrifugation at 14,000 rcf, 10 min, RT. Trypsinization was performed o/n in the dark with a trypsin/protein ratio: 1/100. Tryptic peptides were eluted by centrifugation (14,000 rcf, 10 min, RT). Peptides were dried in a centrifugal vacuum concentrator and stored at −20 °C until LC-MS/MS analysis.

The dried samples were solubilized in a mobile phase A (0.1% formic acid, pH 3.5), sonicated, and peptides were further cleaned up using the SP3 protocol [25]. Finally, the peptide concentration was determined through absorbance at 280 nm measurement on a nanodrop instrument.

### 2.4. LC-MS/MS Analysis

Nano-liquid chromatography of 500 ng tryptic peptide mixture was carried out using a Ultimate3000 RSLC system configured with a 25 cm long pepsep nano column (pepsep.com). The buffer system used was 0.1% (*v*/*v*) formic acid in water (mobile phase A) and 0.1% (*v*/*v*) formic acid in acetonitrile (ACN) (mobile phase B). The flow rate was set to 400 nL/min in the first 10 min of the gradient and lowered to 250 nL/min in the main gradient. The gradient was linear starting from 7% to 35% phase B in 40 min, up to 45% B in 5 min, 45% to 99% B in 0.5 min staying isocratic for 5 min, and then equilibrating at 7% for 10 min at 400 nL/min.

The data acquisition was performed in positive mode using a Q Exactive HF-X Orbitrap mass spectrometer (ThermoFisher Scientific, Waltham, MA, USA). MS data were acquired in a data-dependent strategy selecting up to the top 12 precursors based on precursor abundance in the survey scan (*m*/*z* 350–1500). The resolution of the survey scan was 120,000 (at *m*/*z* 200) with a target value of 3 × 10 × 10^6^ ions and a maximum injection time of 100 ms. HCD MS/MS spectra were acquired with a target value of 1 × 10 × 10^5^ and resolution of 15,000 (at *m*/*z* 200) using a Normalized Collision Energy of 28%. The maximum injection time for MS/MS was 22 ms. Dynamic exclusion was enabled for 30 s after one MS/MS spectra acquisition. The isolation window for MS/MS fragmentation was set to 1.2 *m*/*z*. Two technical replicas were acquired per biological sample.

### 2.5. Data Analysis

#### 2.5.1. Host Proteome Analysis

Using the Proteome Discoverer 2.4, the raw files were searched against the Homo sapiens reference proteome FASTA database downloaded from Uniprot on 19/09/19 (containing protein sequences) using the multiple peptide search (MPS) option activated and SequestHT node. The protein dynamic modifications assessed were oxidation +15.995 Da (M), deamidation +0.984 Da (N, Q), and the N-terminal variable modifications of acetylation +42.011 Da, Met-loss −131.040 Da (M), and Met-loss+Acetyl −89.030 Da (M). Carbamidomethyl/+57.021 Da (C) was set as a static modification. The results were filtered for high confidence peptide identifications, with enhanced peptide and protein annotations. Only master proteins were evaluated. The quantified abundances were based on intensity values and were normalized to the total peptide amount. The statistical evaluation between the control and MIH patient groups was performed using the Proteome Discoverer software pairwise background-based *t*-test, Principal Component Analysis (PCA), and clustering (heat map) functions. The minimum percentage of replicate features (proteins present in one group) was set to 60%. All the identified proteins per sample are listed in Appendix A. Deregulated pathways including differentially expressed proteins are presented in Appendix A. The mass spectrometry proteomics data were deposited to the ProteomeXchange Consortium via the PRIDE partner repository with the dataset identifier PXD035780 [26].

#### 2.5.2. Metaproteomic Analysis

Database search was performed using the MetaLab software (version 1.0), with MaxQuant version 1.5.3.30 involved in the workflow. Carbamidomethyl (C) was set as fixed modifications, and Acetyl (Protein N-term) and Oxidation (M) modifications were included as variable modifications in the database search space. Saliva-specific database downloaded on 5/6-2021 containing 2.859 K sequences (All human oral genomes).

All identified peptide sequences were subjected to taxonomic analysis using the lowest common ancestor (LCA) algorithm implemented in MetaLab (2.2.1). A total of 3882 unique peptides were taxonomically assessed. The calculation of LCA was performed according to the principles implemented in Unipept with modifications. The signal intensity of the peptides was normalized within each sample and expressed in ppm (Appendix A). In order to assess differences in bacterial protein abundance the mean intensity of peptides identified from different taxa was calculated for MIH and control samples along with the ratio MIH/control and *t*-test *p* values.

Data visualization was completed in MEGAN software using the biom file generated from the iMetaLab analysis.

## 3. Results

### 3.1. Clinical Characteristics of the Study Population

Both investigated groups were similar regarding demographic characteristics, dietary habits, fluoride supplement use, plaque index, and gingival index (Table 1).

### 3.2. Proteomic Analysis: Molecular Pathways and Differential Expression of Proteins Involved in Molar-Incisor Hypomineralization

The conducted salivary proteomic analysis generated a comprehensive dataset comprising a total of 1515 high confidence identifications, which is presented in detail in Appendix A. This dataset provides an Atlas of the human saliva proteome both in health and disease. Furthermore, PCA indicates that MIH patients have a significantly different proteomic profile compared to controls as shown in Figure 1. Statistical comparison identified 142 differentially expressed proteins listed in Appendix A. These deregulated proteins are involved in processes such as antioxidant defense, immune response, acute inflammation, and complement activation. Representative proteins that participate in the aforementioned deregulated biological processes are presented in Table 2 with information on their function and expression in MIH compared to controls.

A significant number of deregulated molecular pathways associated with immune response was identified in the MIH group. From the perspective of biological process, the differentially expressed proteins in MIH patients were involved in secretion, as expected in saliva proteome, and in immune response to bacteria. Specifically, based on GO biological processes, acute inflammatory response and factor XII activation were activated. In contrast, innate immune response, defense response to bacteria, and regulation of peptidase activity were repressed in MIH patients. The complete list of deregulated pathways is shown in Figure 2 and detailed information on the proteins participating in each pathway is available in Appendix A.

### 3.3. Microbiome–Microbial Analysis in Molar-Incisor Hypomineralization

Next, we inferred the bacteria species present in each sample through identification of microbial proteins in saliva and assessed differences between MIH patients and controls. Based on the results shown in Appendix A, the number of identified species is lower in MIH samples.

In Appendix A, a heat map of protein intensity originating from different bacteria species is presented and confirms the lower microbial diversity in the MIH group.

## 4. Discussion

### 4.1. Differential Expression of Proteins and Immune Response Deregulation in Patients with MIH

In this study, we performed a characterization of the host proteome and microbiome of MIH saliva and respective controls from healthy adolescents. Our findings show that out of 142 proteins, 30 and 24 proteins were exclusively detected in MIH saliva and control saliva, respectively. Proteins present exclusively in patients’ saliva were functionally linked to “cornification” and “keratinization” with the highest enrichment score. The bioinformatics analysis yields biologically significantly deregulated pathways based on high confidence data.

Differential expression of proteins in the MIH group led to activation of molecular pathways related to pathology, as shown in Appendix A. Regulation of the mechanisms controlling inflammation and synthesis of acute-phase proteins are affected in MIH patients. Additionally, functional defects of the immune system have been correlated with the increased susceptibility of patients to MIH. The pathway analysis indicated deregulation of the critical inflammation and immune response mechanisms in MIH patients. Specifically, enrichment for the biological process revealed that defense response to bacteria, immune response mechanisms, regulation of peptidase activity, and the B-cell receptor signaling pathway were **downregulated** in the MIH group, while leukocyte-mediated immunity, regulation of complement activation, factor XII activation, neutrophil-mediated immunity, neutrophil activation and degranulation, and acute inflammatory response were **upregulated** in the MIH group.

In a similar methodological approach, Bekes et al. analyzed the saliva protein signature in MIH patients, highlighting the role of neutrophils in this pathology and reporting that the proteinaceous composition of saliva is affected in MIH patients, reflecting a catabolic environment, which is linked to inflammation [17]. This analysis supports our findings that the MIH group presents deregulation of pathways related to the immune response, inflammation, and defense response to bacteria. However, it still remains unclear if the neutrophils are only a consequence of the disease and mainly reflect chronic inflammation—or possibly also contribute to its pathogenesis. In their study, Bekes et al. raise the concern of selective absorption of proteins to cotton- or cellulose-based collection devices when saliva is collected using the Salivette method [17,27]. To overcome the concerns and limitations of various collection methods, we used passive drool-collected saliva, which is the method of choice for proteomic analysis [28]. Thus, when comparing the results to other studies, one should consider the intrinsic heterogeneity of individual clinical subjects and the saliva collection method utilized [29].

Our findings agree with Bussaneli et al. [30], who evaluated the association between MIH and polymorphisms in genes encoding essential molecules in the inflammatory response. It is suggested that in addition to polymorphisms in genes directly related to enamel formation, polymorphisms in immune response genes may have a synergistic effect and increase the odds of developing MIH [30]. Alteration in immune response genes may influence the proper development of dental enamel. Furthermore, the interaction between polymorphisms in immune response genes and amelogenesis genes appears to have an additive effect on susceptibility to MIH development in this population [8]. These observations also agree with the current evidence that this disease is not idiopathic and rather multifactorial, with a genetic component related to disturbances during the enamel maturation stage, with the gene-gene and gene-environment interaction being determinant factors for its development [9].

### 4.2. The Oral Microbiome in Patients with MIH

In MIH, the enamel is less hard and more porous compared to healthy controls as it contains a higher content of protein [31]. Unaffected enamel shows a well-organized and distinct prism and crystal structure. In contrast, hypomineralized enamel has less distinct prism edges and crystals, and the interprismatic space is larger [32]. A systematic review by Elhennawy et al. highlights the following characteristics of enamel in MIH patients compared to controls: reduction in mineral quantity and quality (reduced Ca and P content), reduction of hardness and modulus of elasticity (also in the clinically sound-appearing enamel bordering the MIH-lesion), an increase in porosity, carbon/carbonate concentrations, and protein content compared to unaffected enamel [33]. The lower strength of the hypomineralized enamel can result in post-eruptive breakdown soon after tooth eruption or later under the effect of the masticatory forces.

The aforementioned alterations in enamel microstructure consequently facilitate plaque accumulation and development of dental caries. Plaque accumulation is also favored when children with MIH do not brush their teeth due to hypersensitivity of the affected teeth. As expected, due to changes in the local environment, the host–microbe equilibrium is altered. The higher protein content of MIH-affected teeth could favor colonization by proteolytic microorganisms [31,34].

Indeed, a higher protein content in MIH-affected teeth has been reported in previous studies [32,34], where proteins derived from oral fluid and blood, such as serum albumin, indicate that a pre-eruptive disturbance during mineralization can occur, once ameloblasts do not express albumin during amelogenesis [35]. Taking into consideration that cytokines induce potent inflammatory signals [36], it is reasonable to assume that polymorphisms in immune-related genes can alter the expression profile of them, resulting in a greater vascularization of the germ and consequently an increased risk of serum albumin leak. According to Bussaneli et al., future studies should evaluate this hypothesis [30].

Usually, when microbiotas from a healthy and a dysbiotic state are compared, a higher diversity is found in healthy samples, as it happens when comparing healthy teeth surfaces to enamel or dentin caries lesions [37]. Lately, through the study of oral microbiome, dysbiosis is considered a determinant factor of systemic and oral pathology [38]. On a few occasions, the contrary has been observed, such as in periodontal pockets vs. healthy gingival sulci [39]. This has been related to higher nutrient availability or impaired immune system at the affected sites, which would facilitate microbial colonization. In the current study, we hypothesize that deregulation of immune system and defense response to bacteria combined with the porosity of affected enamel alters the host–microbe equilibrium and causes a shift to dysbiosis. In a dysbiotic state, the number and diversity of the microbial species is decreased, as is characteristically presented in the heatmap in Appendix A and in the number of identified bacterial species presented in Appendix A.

Summarizing the above, the results of our study reveal a clear discrimination between MIH and healthy subjects and reflect the deregulation of inflammatory mechanisms in these patients. This deregulation could be related with the decrease in the number and diversity of the microbial species, as presented by the oral microbiome analysis. Thus, our in-depth analysis provides insights into MIH disease pathophysiology and constitutes a basis for future studies, with a focus on non-invasive, therapeutic monitoring of this population.

## 5. Conclusions

This study provides the research community with a high-quality proteomic resource with wealth of information derived from the analysis of host proteome and microbiome, for a very specific patient population. By performing analysis at the systems biology level with rigorous statistical methodology, our research provides functional insights by connecting the disease phenotype to specific biological processes. Regulation of the mechanisms controlling inflammation and defense response to bacteria are affected in MIH patients. Additionally, functional defects of the immune system have been correlated with the increased susceptibility of patients to MIH. Furthermore, bacterial proteome analysis shows a lower diversity for the microbial species, which highlights the dysbiotic environment established in the MIH pathology.

## Figures and Tables

**Figure 1 biomedicines-10-02061-f001:**
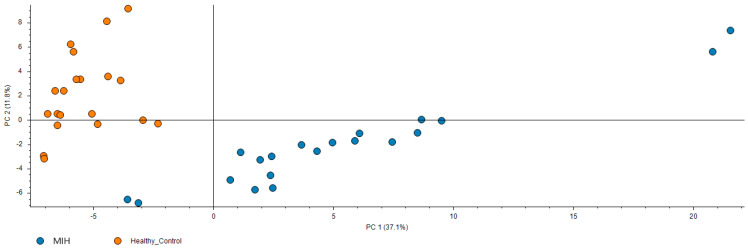
PCA clustering of the MIH (blue dots) and the healthy control (orange dots) reveal a clear discrimination of the two groups.

**Figure 2 biomedicines-10-02061-f002:**
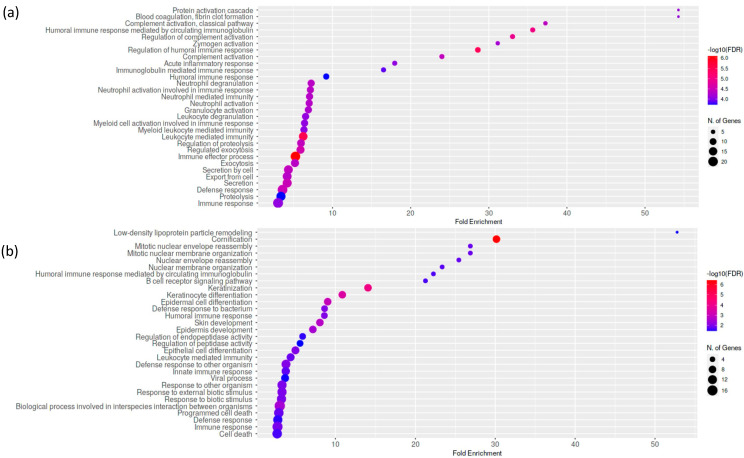
Deregulated biological processes. (**a**) Up in MIH based on GO biological processes. (**b**) Down in MIH based on GO biological processes.

**Table 1 biomedicines-10-02061-t001:** Clinical characteristics and caries index (DMFT) of the studied population (* *p*-value < 0.05). Mean and standard deviation values (means ± SD) are reported.

	MIH	Controls
**Number of Subjects**	10	10
**Age (yrs), mean (SD)**	14.8 ± 1.7	15.2 ± 1.8
**Gender, n (M/F)**	4/6	4/6
**DMFT**	2.3 ± 0.6	2.5 ± 0.4
**Plaque Index (PI)**	0.8 ± 0.05	0.9 ± 0.03
**Gingival Index (GI)**	0.7 ± 0.2	0.8 ± 0.3

**Table 2 biomedicines-10-02061-t002:** Selected deregulated proteins with information on statistical significance (*p* values), expression (ratio MIH/Control), and function.

Protein Name	Ratio MIH/Control	*p* Value	Function
**Immunoglobulin lambda variable 2-11**	2.44	7.12 × 10^−6^	Immune response
**Complement C5**	2.32	2.42 × 10^−5^	Complement cascade
**Plasma kallikrein**	1.67	1.40 × 10^−2^	Inflammatory protease
**Bactericidal permeability-increasing protein**	1.53	4.57 × 10^−2^	Antibacterial activity
**Antileukoproteinase**	0.61	4.25 × 10^−3^	Anti-inflammatory protease inhibitor
**Glutaredoxin-3**	Only in Control	1.00 × 10^−17^	Antioxidant defence
**Glutathione S-transferase Mu 4**	Only in Control	1.00 × 10^−17^	Antioxidant defence

## Data Availability

The MS data presented in this study are openly available and uploaded as Appendix A.

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
