# Peer review of "Analysis of Human and Microbial Salivary Proteomes in Children Offers Insights on the Molecular Pathogenesis of Molar-Incisor Hypomineralization"

_biomedicines, 2022, doi:10.3390/biomedicines10092061_

Round 1

Reviewer 1 Report

The manuscript uses established techniques of proteome analysis to examine differences between salivary proteomes  in patients with and without Molar Incisor Hypomineralization. There have been  some recent uses of these techniques  to examine MIH but the manuscript stands out as clearly written, comprehensive in its scope and offers some interesting insight into MIH and its causes.  The paper is limited somewhat by its broad focus so is not able to draw definitive conclusions but certainly allows the reader to begin to draw some conclusions and points to a number of further areas meriting further study.  The techniques are consistent with those  currently in use in proteome analysis  Again, the paper is extremely  well-written, and I think adds to the literature in this developing area of oral medicine.

Reviewer 2 Report

Dear Authors, 

Thank you for the effort that you put in your work. The topic is really interesting and the work is well structured and explained. 

Hovewer I have some minor concerns,  you will find my comments below.

Introduction

- Introduction  is very well structured and clear. The last paragraph regarding the aim should be probably expanded; e.g. why would the results be useful? 

Material and methods:

- How the 20 subjects were chosen?

- line 7 page 3 - please report the Reference regarding the classification of the EAPD.

Results

- Results are well presented and clear. Line 8 and 9 of page 6: there is a description not fitting any figure / tabel in the text (Suppl Table 1).

Discussion and Conclusion

- Discussion and Conclusions do not give a clear explanation on how the results can be a useful addition and contribution to literature and to the clinical practice. 

- Furthermore there is a lack of a conclusion section! It should be created and conclusions should be clearly presented in all aspects.

I believe once these issues will be overcome, the manuscript will be suitable for publication.

Keep up the good work!
